# Preconception care services in Northern Ethiopia: A qualitative exploration of awareness, experiences, challenges, opportunities, and prospects

Gebremedhin Gebreegziabher Gebretsadik[1,2*], Alemayehu Bayray Kahsay[2], Andargachew Kassa Biratu[4], Amanuel Gessessew[3], Zohra S. Lassi[5,6], Afework Mulugeta[2]

**1** College of Medicine and Health Sciences, Adigrat University, Adigrat, Ethiopia, **2** School of Public Health, College of Health Sciences, Mekelle University, Mekelle, Ethiopia, **3** School of Medicine, College of Health Sciences, Mekelle University, Mekelle, Ethiopia, **4** School of Public Health, College of Medicine and Health Sciences, Hawassa University, Hawassa, Ethiopia, **5** School of Public Health, Faculty of Health and Medical Sciences, University of Adelaide, Australia, **6** Robinson Research Institute, University of Adelaide, Australia

\* gebremedhingebretsad@gmail.com

## Abstract

### Introduction

Preconception care (PCC) has emerged as a key component of the maternal continuum of care worldwide, focusing on reducing poor pregnancy outcomes. Improving services requires addressing opportunities and challenges within the health system, but in Ethiopia, it is often neglected. Hence, this study explores the awareness, experiences, challenges, and opportunities related to PCC services in Tigray, Northern Ethiopia.

### Methods

We conducted an exploratory qualitative study involving 21 in-depth interviews with mothers who experienced adverse pregnancy outcomes and health care providers (HCPs), who work in maternal, neonatal, and child health, and health extension workers. Additionally, we held six focus group discussions with women with a history of pregnancy. We also conducted key informant interviews with 10 maternal, newborn and child health experts from the regional health bureau, district health offices, and professional associations. The study was conducted from January 26, 2024, to April 4, 2024, across four rural districts and two urban areas in Tigray, Northern Ethiopia. Discussions and interviews were audio-recorded, transcribed into the local language "Tigrigna", then translated into English and thematically coded using ATLAS-ti v.7.5.4 software.

**Data availability statement:** Data for this study are publicly available from the Qualitative Data Repository (https://doi.org/10.5064/F6VP6MB9).

**Funding:** The author(s) received no specific funding for this work.

**Competing interests:** The authors have declared that no competing interests exist.

**Abbreviations:** ANC: Ante-natal Care; APOs: Adverse Pregnancy Outcomes; ART: Anti retro-viral therapy; CDC: Centers for Disease Control and Prevention; EPI: Expanded Program Immunization; FP: Family planning; HCPs: Healthcare providers; HEP: Health Extension Program; HEWs: Health Extension Workers; MNCH: Maternal, Neonatal and child health; OPD: Outpatient Department; PCC: Preconception Care; RLP: Reproductive Life Plan; SDGs: Sustainable Development Goals; VCT: Voluntary Counseling and Test; WDGs: Women Development Group; WHO: World Health Organization.

## Results

Some women, particularly those belonging to high-risk groups, are aware of PCC services. Majority of HCPs, especially gynecologists and physicians, have some knowledge of PCC, recognize its importance, and provide specific components of PCC interventions. However, these services are often delivered in a fragmented manner, primarily targeting high-risk women. Identified challenges include traditional beliefs and misconceptions, insufficient counseling on contraceptive services, social influences, service costs, high workloads, lack of medicines and medical equipment, and the fragment-based services . Conversely, opportunities include utilizing existing community platforms and an expressed desire for PCC services. Moreover, diverse communication strategies, linking communities with health facilities, involving high-risk mothers as educational role models, and integrating package-based PCC services into the healthcare system were explored as perceived suggestions.

## Conclusion

Apart from high-risk women, most women have little to no awareness about PCC services. Furthermore, although many HCPs possess some understanding of PCC, they deliver only a limited range of interventions, primarily catering to self-initiated high-risk mothers. Challenges identified include traditional beliefs and misconceptions, inadequate counseling on contraceptive services, social influences, high service costs, and fragmented service delivery. Existing community platforms and the perceived desire for PCC services were highlighted as opportunities to enhance PCC services. Strategies such as utilizing diverse communication methods, involving high-risk mothers as role models, strengthening community engagement activities, and improving linkages between communities and health facilities were proposed. Additionally, promoting home-based self-care was explored as a suggestion for improving PCC services. Integrating package-based PCC services into the healthcare system to routinely serve all eligible women of reproductive age was recommended to improve both awareness and uptake of PCC. Finally, tailored interventions were deemed essential for improving PCC awareness and utilization both at the community and facility levels.

## Introduction

Preconception care (PCC) has garnered significant attention globally, primarily due to its ability to reduce the risk of adverse pregnancy outcomes (APOs). The existing continuum of care lacks pre-pregnancy care, and PCC addresses this gap by mitigating parental risk factors before conception, thereby enhancing outcomes for both mothers and infants [1]. To mitigate the burden of APOs, both the World Health Organization (WHO) and the Centers for Disease Control and Prevention (CDC) universally recommend PCC as an essential part of maternal healthcare. They advise that

women intending to become pregnant should receive at least one preconception care checkup [2]. Following CDC/WHO recommendations, several low- and middle-income countries (LMICs), including Bangladesh, the Philippines, and Sri Lanka have innovatively integrated PCC into their health systems [2,3]. Besides, some Sub-Saharan Africa(SSA) countries, like South Africa [4], Kenya [5], and Ethiopia [6,7], have also incorporated PCC into their healthcare systems and are working to leverage this initiative to achieve Sustainable Development Goals (SDGs) on time.

Worldwide, approximately 295,000 maternal deaths occur each year due to complications from pregnancy or childbirth [8]. Additionally, there are around 23 million miscarriages [9], 14.8 million live preterm births [10], and 295,000 neonatal deaths caused by congenital anomalies annually [11]. The prevalence of congenital anomalies in LMICs is significantly high, with 94% of cases being severe [12]. Similarly, the burden of APOs in LMICs remains a significant global issue. For example, the burden of APOs in SSA is 29.7% [13]. Ethiopia is among the LMICs with high rates of maternal and neonatal deaths, at 412 per 100,000 live births and 33 per 1000 live births, respectively [14,15]. Before the war, Tigray's health facilities ranked among the best in Ethiopia for maternal, newborn, and child health services [16]. In the Tigray region, neural tube defects occur at a rate of 131 per 10,000 births [17]. However, since the conflict began in November 2020, significant damage to the health system has severely disrupted maternal care and essential services. This has exacerbated preconception risks, including unwanted pregnancies, unsafe abortions, home deliveries, malnutrition, and increased maternal, newborn, infant, and child mortality rates [18–20]. For instance, the maternal mortality rate during the conflict reached 840 deaths per 100,000 live births [21].

Despite the WHO's recognition of its feasibility in LMICs, PCC has not been successfully adopted in Africa [2]. HIV testing, family planning, and contraceptive services are commonly utilized components in PCC interventions, while supplementation of folic acid is less widely used [22]. A recent review in SSA found that the uptake and knowledge of PCC were only 24.05% and 33.27% [23], respectively. A systematic review further revealed that in Ethiopia, only 30.95% of women knew about PCC, with just 16.27% utilizing it [24], and the provision was low at 15% [25]. Ethiopia has set ambitious targets under its reproductive health strategic plan to align with SDG 3.1, aiming to significantly reduce preventable pregnancy-related morbidity and mortality by 2025 [26]. These targets include increasing the proportion of pregnant women receiving PCC to 25%, reducing the neonatal mortality rate from 33 to 21, and decreasing the maternal mortality rate to 271 per 100,000 live births [26]. However, information is lacking on how the healthcare system has implemented PCC services.

Studies have indicated that identifying practical and feasible interventions based on local contexts is essential for raising awareness and improving PCC service adoption [27,28]. A qualitative study in high-income countries revealed that unfavorable attitudes, a lack of knowledge about PCC, limited resources such as time and guidelines, and unclear responsibilities for providing PCC are barriers to its provision [29]. In Southwest Ethiopia, a study focused on barriers to the uptake of PCC identified some barriers, including poor knowledge, unplanned pregnancies, heavy workloads, service costs, distance, unavailability of services, and insufficient attention from media personnel [22].

Although Ethiopia has strategically integrated PCC into its health system with newly developed guidelines [6,7], there is limited evidence, particularly on the challenges and opportunities within the system that influence service enhancement. This study addresses this gap by examining awareness, experiences, challenges, and opportunities from the perspectives of experts, healthcare providers (HCPs), and women with pregnancy histories. It evaluates current practices and provides recommendations to improve PCC awareness and uptake, guiding the development and implementation of effective interventions.

## Materials and methods

### Study design

We employed an exploratory qualitative study design to investigate awareness, experience of the current practices, challenges, and opportunities for PCC services. The study was reported in accordance with the Consolidated Criteria for Reporting Qualitative Research (COREQ) checklist [30] (S1 File COREQ Checklist for Study Reporting).

## Study setting

We conducted the study in the eastern and central regions of Tigray, Northern Ethiopia, with populations of 994,346 and 1,522,596, respectively. According to the Tigray Regional Health Bureau, in 2020, the region had a total of 14,423,731 HCPs, including 3,074 health extension workers (HEWs). These zones include 591,481 women of reproductive age (23.5% of the population). The study focused on specific districts and urban areas in both zones from January 26, 2024, to April 4, 2024. We specifically targeted two rural districts and one urban district in the eastern zone, along with two rural districts and one urban woreda in the central zone of Tigray. The region invested in primary health care units, achieving a 91.7% coverage rate. However, the war-damaged over 80% of health facilities, leading to a 40% decline in maternal and child health services, including institutional deliveries [31].

## Participants

In the focus group discussions (FGDs), we included women who have had a history of pregnancy or are currently pregnant and who have the intention to be pregnant. Mothers intending to become pregnant were identified using a single, widely accepted reproductive life plan (RLP) tool [32] and were reached through HEWs and Women's Development Groups (WDGs) in the community. We recruited high-risk mothers (history of APOs like stillbirth, neonatal death, congenital anomalies, perinatal death, miscarriage, post-partum hemorrhage, recurrent abortion [33] or history of chronic medical diseases like diabetic mellitus, hypertension, or HIV) for the in-depth interviews (IDIs). Using purposive sampling, we identified mothers from HEW registers who had engaged with HEWs and WDGs, based on pregnancy status and risk factors. In the Ethiopian context, HEWs work across health posts, communities, and households in coordination with WDGs. Furthermore, HCPs, including midwives, nurses, health officers, and medical doctors stationed in Maternal, Newborn, and Child Health (MNCH) units, were identified with the guidance of the medical director or MNCH coordinators of these health facilities for IDIs. Additionally, experts from the District Health Office of MNCH or Health Extension Program (HEP) case team in selected districts, the Reproductive Maternal Neonatal Child Health (RMNCH) case team from the regional health bureau, and professionals from the Ethiopian Midwives Association (EMA) and Ethiopian Obstetrics and Gynecology Association (ESOGA) participated in the Key informant interviews (KIIs). Furthermore, we selected HCPs for both IDIs and KIIs based on their substantial experience in MNCH and their ability to communicate effectively, thereby providing valuable insights into the current state of PCC.

## Data collection

We obtained participant information through IDIs, KIIs, and FGDs. To triangulate and validate the data, we included mothers, medical professionals working in MNCH units, MNCH experts from district and regional health offices, and associations.

Mothers for FGDs and IDIs were identified with the assistance of HEWs and WDGs within the community at their households. After obtaining their consent, we interviewed them in private settings, such as village health posts or households, to ensure privacy and minimize background noise. Medical directors supported selecting HCPs for IDIs, who were then interviewed in private rooms at their workplaces. Similarly, we conducted face-to-face interviews with MCH experts for KIIs in private rooms at their workplaces. The number of participants was determined based on data saturation, which occurs when participants' descriptions become repetitive. Sampling continued until no new information emerged, indicating that saturation was reached [34].

Six FGDs were conducted, one in each district, involving 7–9 mothers to become pregnant, each session lasting 56–96 minutes. Eight mothers with a history of APOs participated in IDIs, averaging 42 minutes per interview. Additionally, 13 HCPs from six health centers and health posts were interviewed, averaging 46 minutes each. KIIs involved 10 MNCH experts, seven from district health offices and RHB, and three clinicians and academics from ESOGA & EMA, with 38–70 minutes of interviews.

The primary investigator (GG) and three other PhD students in public health (GB, KK, and FT) who are experienced and trained in qualitative research conducted interviews and FGDs. The four interviewers were paired into two groups: one note-taker and one interviewer. We used interviews and discussion guides to explore experiences, challenges, and opportunities in providing PCC services. We developed semi-structured guides for four groups: mothers to be pregnant, high-risk mothers, HCPs working in the MNCH unit, MNCH experts, clinicians, and academics in MNCH. These guides were initially drafted in English and translated into the local language, Tigrigna. FGDs were organized in circular seating arrangements to facilitate interactive discussions among participants [35]. The interview guides comprised open-ended questions [supplementary S2 File]. Furthermore, the interview guide incorporated probing questions. Interviewers actively guided respondents through the questions outlined in the guide, using probing techniques to prompt further explanation of participants' responses. Throughout the interviews, they audio-recorded all sessions, took field notes to document key points, and observed participants' non-verbal cues in-depth.

## Trustworthiness

We ensured the reliability of our findings by implementing rigorous quality measures. The data collectors were trained in tools, interview techniques, participant selection, the concept of reflexivity, and consent procedures. We pretested the interview guide in a similar setting before starting data collection and revisions were made based on feedback. During data collection, the team held daily debriefing sessions to address emerging issues and spent extended time with participants to gain deeper insights. We extended the research period to gain an in-depth understanding of the phenomena. We shared participant transcripts for verification and incorporated their feedback. We conducted data collection and analysis simultaneously, triangulating findings from interview transcripts with field notes. Experienced researchers fluent in the local language and culture translated, transcribed and coded parts of the audio recordings while the primary investigator reviewed their work to ensure accuracy. Preliminary results were presented to experts and peers, further refining the guides. While conducting participant interviews, creating codes, and organizing these codes into categories and themes, the research team bracketed their prior experiences and knowledge to enhance the quality of the results. To ensure the consistency of our findings, we conducted member checking with four participants, each representing a distinct group: HCPs, MNCH experts, and the two groups of mothers. Additionally, we employed IDIs, KIIs, and FGDs as data collection methods to inform and guide subsequent discussions.

## Data analysis

The FGDs, IDIs, and KII were audio-recorded, transcribed verbatim in the local language "Tigrigna", and then translated into English. Verbatim transcripts of the data saved as an independent MS Word file were imported, stored, managed, and coded using Atlas. ti v.7.5.4 (Scientific Software Development GmbH, Berlin, Germany) qualitative data analysis software. The first author (GG) conducted the initial coding by thoroughly listening to the audio recordings from the FGDs and interviews and carefully reviewing all the transcripts to become familiar with the content and to identify initial coding. In addition, we linked field notes and investigator memos to the respective files in the software to assist in the analysis.

The first author (GG) conducted line-by-line coding. We used both process and values coding to analyze all the transcripts. Additionally, we applied a hybrid approach, inductive and deductive coding methods during the analysis [36]. Another investigator (AM) coded the five interviews and the FGDs to check inter-coder reliability. Transcription, translation, analysis, and data collection were conducted simultaneously throughout the data collection period. The investigators grouped similar codes to create categories and subcategories. We reviewed and revised the codes for clarity and consistency, then consolidated them to eliminate redundancy and overlap in the analysis. We used a thematic analysis approach to identify the major themes across the categories and subcategories [37]. Furthermore, during the write-up, we performed content analysis to describe the frequency of participants in subcategories. Quotes that best described the various categories and frequently expressed sentiments across several groups were chosen and presented in italics.

### Ethics approval

The Institutional Review Board of Mekelle University, College of Health Sciences (reference: MU-IRB2075/2023) granted ethical approval. The Tigray Health Bureau issued a support letter, and the respective district health offices and villages granted permission. We fully explained the study's objectives, risks, and benefits and obtained informed consent from all participants. The college's institutional review board approved the consent form. We prioritized privacy and confidentiality and informed participants about their right to withdraw at any time. We also requested permission to record focus group discussions and interviews.

## Results

### Socio-demographic characteristics of the respondents

The qualitative study involved 79 participants, including 21 individuals in IDIs with HCPs and high-risk mothers. Additionally, 10 MNCH experts from district health offices, regional health bureaus, and professional associations participated in KIIs. Among the high-risk mothers, 63% were aged between 18 and 34 years, all were housewives, and 12.5% were grand multiparous. Of the healthcare professional participants, 64% were female, with approximately 74% being midwives, public health professionals, or HEWs (Table 1). Additionally, 48 mothers who become pregnant in the future participated in the FGDs. Among these FGD participants, 63% were between 18 and 34 years old, and 46% had an educational level of high school or above (Table 2).

The conceptual framework, which emerged from qualitative data, illustrates the overall relationships among the factors involved in PCC service. A higher level of awareness and positive experiences positively influences PCC services. Conversely, challenges such as traditional beliefs, misconceptions, and fragmented PCC services negatively affect PCC services, including the level of awareness and experiences. This framework) provides a guiding structure for implementing PCC services within the healthcare system (Fig 1)

1. Awareness of PCC services

### Information on PCC

Access to information about a service is essential for utilizing it effectively. Without awareness of its existence, people cannot benefit from it. For many women and community members, the concept of PCC remains unfamiliar, as it is a relatively new approach that the healthcare system has not actively promoted. Both mothers and HCPs indicated that most women lack information about PCC, turning to healthcare services only after pregnancy confirmation or fertility difficulties. This gap in understanding repeatedly results in confusion between PCC and prenatal care. High-risk mothers stated:

…..*"I do not know what pre-pregnancy care is?"* **(32 years old mother, 5ᵗʰ grade, IDI)**

."*Endie….the meaning of this local language is* **(***I don't know*)**. I am just keeping silent" **(28 years old mother, 8ᵗʰ grade, IDI)**

Similarly, a Clinical midwifery professional said,

"*…..there is a lack of information regarding preconception in the community; I couldn't expect mothers to visit and utilize preconception care. I believe that we healthcare providers and health professional associations didn't invest in it, and from my observation in our context, I can conclude that no mothers are coming to preconception care, even though it is not supported by research*" **(Midwifery professional, male, KII)**

On the Other hand, participants noted that some high-risk mothers have information about the services. They emphasized that HCPs should advise these mothers, including those with a history of diabetes, hypertension, HIV, spontaneous abortion, infertility, or congenital abnormalities, to undergo various health screenings before conception. These screenings

Table 1. Characteristics of the study participants in Tigray, Northern Ethiopia, 2024.

| Characteristics | | IDIs(N=21) | | KIIs(N=10) |
|---|---|---|---|---|
| | | High risk mothers* (n=8) | HCPs* (n=13) | HCP experts* |
| Sex | Male | NA* | 3 | 5 |
| | Female | | 10 | 5 |
| Age of participants, years | 18-34 | 5 | 8 | 1 |
| | 35–49 | 3 | 5 | 7 |
| | =>50 | 0 | 0 | 2 |
| Work experience, years | <5 years | NA | 2 | 1 |
| | 5-10 years | | 6 | 5 |
| | 11-20years | | 5 | 4 |
| Occupation | Housewife | 8 | NA* | NA* |
| Educational level | Not attended school | 1 | NA* | NA* |
| | Primary (1–8th | 3 | | |
| | Secondary(9–12th) | 4 | | |
| | Diploma | 0 | 5 | 0 |
| | BSc. Degree/MD* | 0 | 8 | 5 |
| | MSc. Master/specialty | 0 | 0 | 5 |
| Profession | Midwifery | NA* | 5 | 2 |
| | Public Health | | 1 | 5 |
| | Family health | | 0 | 2 |
| | Nursing | | 2 | 0 |
| | MD/specialty | | 1 | 1 |
| | Health Extension Workers | | 4 | 0 |
| Current working unit | Clinician & academic | NA* | 0 | 3 |
| | MNCH expert | | 0 | 7 |
| | MNCH unit | | 9 | 0 |
| | HEP* | | 4 | 0 |
| Number of births | 0-1 | 2 | NA* | NA* |
| | 2-4 | 5 | | |
| | =>5 | 1 | | |

NB: *PCC: Preconception Care, HEP: Health extension program, MD: Medical Doctor, MNCH: Maternal, Neonatal, and Child Health, HCPs: Healthcare Providers working in Maternal, Neonatal, and Child Health related units, NA: Not Applicable: HCP Experts: Healthcare professionals working at woreda health offices, regional health bureaus, and academic institutions: High-risk mother: a mother with a history of adverse pregnancy outcomes or chronic medical conditions.*

should include tests for non-communicable and communicable diseases, sexually transmitted infections, mental health issues, gender-based violence, and other chronic conditions. Additionally, they should receive advice on proper nutrition and avoid harmful substances like alcohol. A high-risk mother said,

*I have heard information on PCC from media outlets, such as Tigrai Television, in a program called "Maeda Hakaym." I think it is good and beneficial to have a planned life and children. It sounds nice to have plans, preparations, and examinations and give birth based on planned decisions. Screening for gender-based violence, chronic disease, sexually transmitted infections, substance use, mental health, nutritional counseling and balanced diet and consultation to a doctor in the presence of these factors and medication use is essential to prevent adverse pregnancy outcomes"* **(35 years old mother, 10 th grade, IDI).**

Even some mothers explain the concept of PCC and understand its benefits.

**Table 2. Characteristics of FGD participants in Tigray, Northern Ethiopia, 2024.**

| Characteristics | | (FGD* 1–6) =48 mothers |
|---|---|---|
| Age of participants, years | 18-34 | 30 |
| | 35–49 | 18 |
| Occupation | Housewife | 41 |
| | Student | 2 |
| | Merchant | 4 |
| | Daily laborer | 1 |
| Educational level | Not attended school | 3 |
| | Primary (1–8th) | 23 |
| | Secondary(9–12th) | 13 |
| | Diploma | 8 |
| | Degree | 1 |
| Number of births | 0-1 | 14 |
| | 2-4 | 30 |
| | =>>5 | 4 |

NB: FGDs*, Focused Group Discussion.

**Themes:** Through an analysis of the transcribed interviews, five major themes, encompassing nineteen sub-themes, emerged. Awareness, experience, challenges, opportunities, and suggestions have emerged from the qualitative data on PCC services (Table 3).

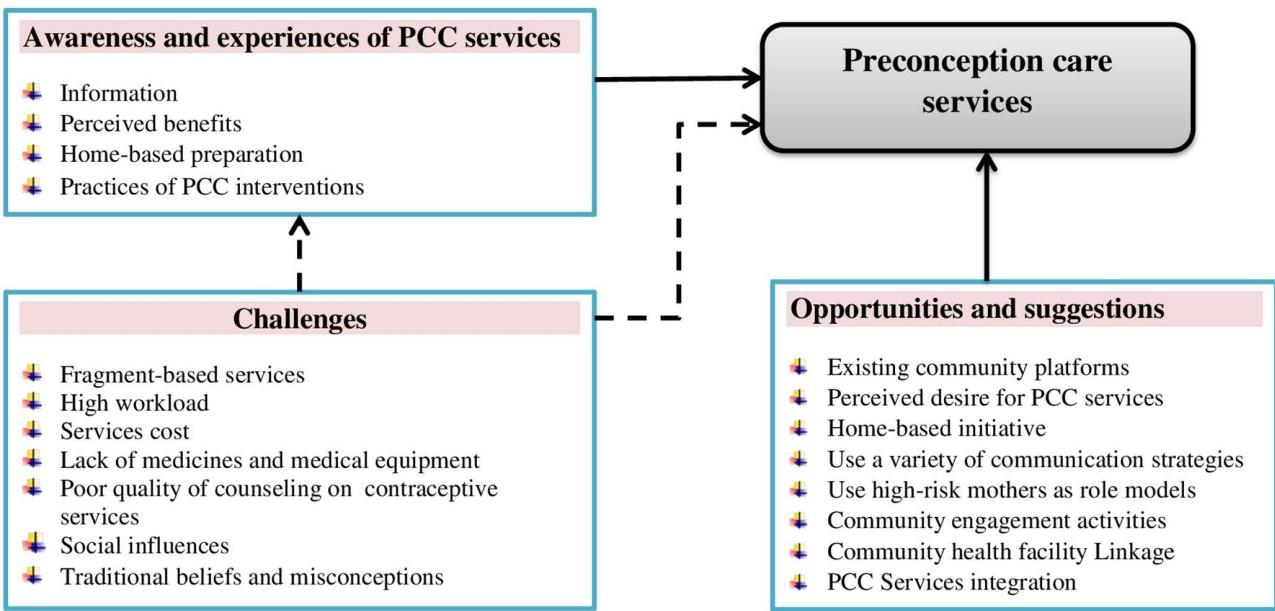

**Fig 1. Framework of Awareness, Experiences, Challenges, and Opportunities of PCC Services in Tigray, Ethiopia.** Note: The dashed lines represent a negative influence on PCC services, while the solid lines indicate a positive influence.

*"I understand that PCC means I need to check my health status, should not take alcoholic drinks, need to have mental stability and limit the number of children that I have according to my income or relation to livelihood. These come to my mind when I am thinking about PCC"* **(29 years old mother, 10th grade, IDI)**

**Table 3. A list of themes and sub-themes emerged from the data, Tigray, Northern Ethiopia, 2024.**

| Major themes | Sub-themes |
|---|---|
| Awareness of PCC services | Information |
| | Perceived benefits |
| Experiences of PCC services | Practices of PCC interventions |
| | Home-based preparation |
| Challenges of PCC services | Fragment based services |
| | Traditional beliefs and misconceptions |
| | Poor quality of counseling on contraceptive services |
| | Social influences |
| | High workload |
| | Services cost |
| | Lack of medicines and medical equipment |
| Opportunities for PCC services | Existing community platforms |
| | Perceived desire for PCC services |
| Suggestions for PCC services | Home-based initiative |
| | Use a variety of communication strategies |
| | Use high-risk mothers as role model |
| | Community engagement |
| | Community -health facility linkage |
| | PCC services integration |

Besides, most HCPs have some knowledge of PCC and emphasize the importance of women's health before conception. Gynecology professionals, medical doctors, and, to a lesser extent, midwifery professionals have a better understanding of PCC concepts and interventions. Key PCC services identified by HCPs include HIV testing, folic acid supplementation, nutritional education, family planning, chronic disease screening, substance use advice, STI/HIV screening, medication management, and assessment of APOs such as Rh incompatibility, Td vaccination, abortion, and congenital anomalies. Among these, folic acid supplementation, nutritional advice, and family planning were most frequently mentioned.

*HIV testing, folic acid supplementation, feeding practices, and nutrition are the things I know about preconception care services* **(Midwifery professional, female, IDI)**

All participants concurred that nearly all HCPs are knowledgeable about recommending three months of folic acid supplementation to mothers with a history of congenital anomalies and repeated abortions before conception.

*Mothers who had a history of repeated abortion before conception are eligible to take folic acid for three months. We don't give folic acid supplementation to all eligible women* **(HEW, female, IDI)**

On the other hand, participants reported that some HCPs have no information about PCC and do not even recognize its name.

*……. No, I don't know. I don't even think there is such care here. It has been a long time since I started working here; I never heard anyone talking about it, and I never saw such service being provided here* (**Nursing professional, female, IDI**)

*We don't even know about preconception care ourselves. We health professionals don't know it* (**HEW, female, IDI**)

### Perceived benefit of PCC services

Mothers and HCPs emphasized that PCC offers valuable opportunities to assess women's health, enabling early identification and management of potential issues. This approach not only enhances maternal healthcare services but also helps avoid unnecessary expenses and reduces the risk of mother-to-child HIV transmission.

**Give opportunities for screening of health status**

PCC provides opportunities to assess the health status of mothers, enabling early detection of various conditions and diseases and identifying potential preconception risks that may lead to adverse pregnancy outcomes, as noted by participants. This aspect contributes to the promotion of women's health.

*"Preconception care plays a crucial role in screening for various health risks such as substance use, gender-based violence, mental illness, HIV/STIs, and chronic conditions like diabetes, hypertension, and heart disease before conception occurs"* **(29 years old mother, 10th grade, IDI)**

**Save unnecessary expenses and efforts.** Participants pointed out that some women spend their money and energy on ultrasounds and other laboratory tests privately during pregnancy because they did not consult HCPs before becoming pregnant. However, if these tests were initiated earlier, mothers could avoid unnecessary expenses and efforts.

2. Experience of PCC services

**Current Practices in the Components of PCC interventions**

HCPs highlighted a significant gap in healthcare settings, attributing it to the absence of routine PCC services with designated providers or specialized units, leading to missed opportunities for comprehensive care. They observed that PCC interventions, such as folic acid supplementation and counseling, are delivered in a fragmented manner across various units, including antenatal care, gynecology/obstetrics, youth-friendly services, family planning, chronic disease clinics, and post-abortion care, particularly for high-risk mothers. Although midwives and gynecologists provide some services, participants emphasized that private clinics offer them more consistently.

A public health professional discussed how various units provide components of PCC interventions: *"It is done irregularly in youth-friendly services units like the family planning itself, reproductive and diet education. But it is not deep. But something is tried. Usually, as I tell you, the peer-to-peer takes the biggest share"* **(Public health professional, male, KII)**

*…..in addition to these, mothers diagnosed with DM, threatened abortion, spontaneous abortion, recurrent stillbirth etc, are informed and appointed to consult a healthcare provider before getting pregnant in gyn/obs OPD. The consultation is specifically regarding folic acid; there is no more service provided for other possible reasons* **(Midwifery professional, male, KII)**

In general, most participants noted that the components of PCC interventions provided mainly for high-risk women are:

**Contraceptive counseling for pregnancy delay**

All participants, including both users and providers, noted that contraceptive counseling for delaying pregnancy was offered to all women of reproductive age within the community and healthcare system.

**Provide folic acid supplementation**

All participants, both users and providers, agreed that folic acid supplementation was commonly provided to mothers with a history of chronic diseases such as diabetes, congenital anomalies, and spontaneous abortion, making it one of the most frequently practiced components of PCC interventions.

*……"We provide folic acid for women who have a history of abnormal pregnancy like spinal bifida, hydrocephalus, and anencephaly. We give priority to these women"* **(MD, male, KII)**

**Counseling about medication safety**

Participants mentioned that some women with a history of chronic diseases, such as diabetes and hypertension, received counseling about medication safety before conception when they visited the health facility for follow-up.

" *Need advice about medication with alcohol, if mothers have diabetic mellitus, HIV, and Hypertension, should advise her that the medicines should be changed first before pregnancy,for example, ACE inhibitor drugs have teratogenic effects*" *(MD, male, IDI)*

**Counseling about substance use.** Some mothers with a history of APOs, including abortion, chronic disease, congenital anomalies, and alcohol intake, received suboptimal counseling. They were advised to avoid medications that cause teratogenic effects and to abstain from substance use. Additionally, they were recommended to take folic acid supplements and undergo HIV testing before conception.

*Women are counseled to take folic acid before 3 months of conception and to avoid bad habits such as alcohol, smoking, and other addictive substances* **(Family health professional, female, KII).**

Contrarily, an MNCH expert employed at the district health office provided insight into the availability of PCC services within healthcare facilities as follows:

*In both urban and rural locations, pre-pregnancy care is currently nonexistent. It didn't exist before, as far as I can tell. Policy, too, begins with family planning. Additionally, access to preconception care is not as high as indicated. I won't be questioned about whether I offer the services. I'm speaking to you based on my level of expertise. It was recently that I attended training. I received thorough training on PCC in Ethiopia,* **XXX town,** *and it isn't offered locally* **(Public health professional, male, KII)**

Moreover, most of the mothers agreed that women seek pre-pregnancy care only when concerned about their health. A high-risk mother noted:

*As a result of my personal health issues, I sought advice from healthcare providers. I was diagnosed with hypertension, which unfortunately led to a stillbirth in the past. Therefore, I consulted with health workers to ensure the best possible outcome when I planned my next pregnancy. I even visited referral hospitals for further guidance and was advised to visit healthcare facilities ahead of any future pregnancy. I followed this advice accordingly* **(35 years old mother, 9th grade, FGD)**

Additionally, mothers reported not receiving any counseling as part of pre-pregnancy care, even those who visited health facilities to have their contraceptives removed in preparation for pregnancy.

*"When you go to health facilities for contraceptive removal, the care providers will ask you, 'Is it because you want to conceive? "Then, they would simply remove you and do nothing else, even if you said yes. There is no preconception care, there is no counseling, and they didn't assess your eligibility for pregnancy. Due to this, the community doesn't have the awareness, doesn't know what PCC is or what to do, and as a result, there is no demand for the service."* **(35 years old mother, 10th grade, IDI)**

**Home-based preparation**

Adopting a pregnancy plan enables women to prepare effectively for conception, reducing modifiable risks and promoting healthy pregnancies and positive birth outcomes. While most mothers lack access to PCC services, some, especially high-risk mothers, take proactive steps at home, such as saving money for medications, modifying alcohol intake, improving their diet, and focusing on mental readiness.

*"Before the mother becomes pregnant, she needs to get prepared psychologically and economically. Psychologically prepared means mentally she needs to be stable, she needs to eat more than usual (4-5 times per day)"* **(29 years old mother, 10th grade, IDI)**

A mother mentioned that when planning for pregnancy, she prepares the chicken with local alcohol (siwa) for her husband and drinks " siwa tsiray" herself, believing it benefits the child.

*…….." A chicken should be slaughtered and Siwa (local alcohol) should be prepared, it is done before you get pregnant or when you are thinking about pregnancy. It is mostly for my husband, but I also drink the "siwa tsiray", and it is good for the child"* **(35 years old mother, not attended school, FGD)**

On the contrary, participants emphasized that most mothers, particularly those without known health issues, do not take any action or invest in pregnancy preparation. A high-risk mother stated:

*……"I didn't plan to get pregnant. In my situation, I did not take any extra precautions before becoming pregnant, so I continued to eat normally and did not seek medical advice or counsel on PCC until after I became pregnant. **(29 years old mother, 10ᵗʰ grade, IDI**)*

3. Challenges **of** PCC services

**Fragment-based services.** Participants observed that HCPS deliver fragmented PCC components to high-risk women, resulting in low awareness and uptake. Key challenges include the lack of PCC guidelines, limited government focus, and inadequate training.

*"In general, the health system gave no focus to PCC. In practice, healthcare institutions have established no structures to support PCC. I think this is the main challenge. Hence, if there is no center for PCC in the health system, if there is no focus on PCC, and if the health care provider does not emphasize PCC, the community will do so. I believe this is the primary challenge" **(MD, male, KII)***

All participants noted that while limited orientation or information about PCC was provided, the government prioritized and delivered package-based services such as antenatal care (ANC) and delivery services.

*"A kind of orientation or information was given about PCC by partners, but they still do not get that much focus from the government or partners on PCC, like the other health care services. Through education, the community shows behavioral change about child and maternal nutrition" **(Family health professional, female, KII)***

**Traditional beliefs and misconceptions**

**Lack of felt need to disclose desire to conceive.** Mothers and HCPs have noted that, with few exceptions, most women tend to keep their desire to have a child confidential. Many women even conceal their pregnancies during the early months. They typically do not discuss their intentions to conceive with others unless it is with their husband, a very close friend. While discussing pre-pregnancy care may seem taboo or embarrassing within specific segments of the community, it might be disclosed after pregnancy is confirmed because pregnancy is perceived as a gift from St. Mary or God.

A high-risk mother said:

*………" Discussing care before pregnancy with a neighbor or a part of the community is often considered impolite or embarrassing. Instead, it's more common to share this information once pregnancy is officially confirmed, as there's a belief that pregnancy is a blessing from St. Mary or God **(32 years old mother, 5th grade, IDI**)*

*"Mostly, it is in the latter stages of pregnancy that the woman discloses her pregnancy status; except for a few, most of the women keep it confidential" **(29 years old mother, 10ᵗʰ grade, IDI**)*

Additionally, an MCH expert said: *"Disclosing or consulting with healthcare providers before getting pregnant is the biggest challenge in the community. Mothers may feel ashamed to consult; normally, it shouldn't be "**(Public health professional, female, KII)***

**Misconception about fear of side effects.** Participants noted that contraception, especially the Depo injection, is believed to cause infertility. Women avoid using Depo because it delays pregnancy after discontinuation. Due to this misconception, many women are discouraged or restricted from using contraceptives.

*"Now, for example, faith in the contraceptive (depo) causes a delay of pregnancy and hence many mothers do not use contraception" **(MD, male, IDI)***

A maternal health expert pointed out that mothers discontinue contraceptives because they believe these may be responsible for the increase in congenital anomalies.

*……" You see, especially now, after encountering many congenital anomalies, numerous mothers have approached us to discontinue family planning" **(Midwifery professional, female, IDI)***

**Misconception about intake of alcohol.** Participants noted that consuming traditional alcohol ("Siwa" and "Myes") is not harmful except when taken with medication, particularly during the first four months of pregnancy. Additionally, some mothers believe that alcohol does not increase the risk of abortion or stillbirth; instead, they attribute these risks to poor nutrition, lack of adequate rest, and stress. Participants also mentioned that some elders believe consuming honey before pregnancy may facilitate the mental development and good health of the fetus later during pregnancy.

*"There is no problem with the intake of alcohol before pregnancy. Even though it does not cause any harm until four months of pregnancy, according to the lesson we took, drinking alcohol is not allowed when you are taking medication. However, there is no problem with other issues. Drinking milk is not allowed after six months of pregnancy"* **(35 years old mother, not attended school, IDI**)

*"Yes, for instance, our elders say it is good to take honey before pregnancy. They say it facilitates mental development and the good health of the foetus later during pregnancy. Also recommended by the community is "Myes" intake. I don't think these substances can cause abortion or stillbirth. Abortion and stillbirth are mainly caused by the factors I mentioned earlier, such as poor nutrition, a lack of adequate rest, and stress"* **(35 years old mother, 10th grade, IDI.**

### Poor quality of counseling on contraceptive services

Mothers and HCPs agreed that inadequate counseling often deters women from using contraceptives, as they worry about side effects like delayed conception and bleeding. When women discontinue contraceptives, switch to alternatives, or stop using them altogether, unintended pregnancies can occur, disrupting the effective delivery of PCC services.

*"I was on birth control for four years, but stopped because I experienced intense bleeding or a hemorrhagic condition, which led to my pregnancy. It wasn't something I had planned for. I only used contraceptives for the first pregnancy (since it was my first, I didn't use them again), but all my other pregnancies were unplanned. They happened after I stopped using birth control due to negative side effects. I attempted to use birth control again but stopped due to these side effects, and tried the calendar method, but that didn't work, and I ended up pregnant again. I didn't have a specific plan for my reproductive life. A lot of it just happened by chance, not by design."* **(35 years old mother, 10th grade, IDI)**

*"The healthcare provider is also sending mothers for each contraceptive without properly explaining what is wrong with it and what kind of mother it should be given to"* **(MD, male, IDI)**

### Social influences

Participants highlighted the influential role of husbands, older in-laws, and community leaders in encouraging women to plan pregnancies and seek healthcare advice. However, cultural beliefs often dictate that once a woman is pregnant, she must continue the pregnancy, regardless of her mental readiness. In some areas, women fear abandonment if they leave their husbands during pregnancy, while cultural norms discourage open discussions about family planning with in-laws or friends, causing embarrassment. Additionally, some elders advocate seeking blessings from holy water instead of consulting healthcare professionals.

*"What is the barrier to pre-pregnancy testing that you explained is the backward culture. Even though she knows it, she wants this; she wants to have a baby because she thinks it will hurt, or she is ashamed, it's a shame to be told this is what she wants, but they know it. It is very embarrassing, not very comfortable to say that you want to have children with your mother-in-law or father-in-law, even with your friends. For example, I can tell them that I want to have a baby here, and my baby is grown. I may even have a desire inside, but they can go out and accuse me of being shameful"* **(35 years old mother, 10th grade, FGD)**

*Some would believe in the service, and some would not, especially the old people (elders).*

*The elders may downplay the service like: "we are here and still managed to have kids despite there being no such service in our time," and ask, "Consulting who else were our grandmothers conceiving that you are doing so now?" and the like* **(35 years old mother, 10th grade, IDI)**

Mothers and HCPs observed that while some husbands with higher literacy levels recognize the importance of PCC, the majority perceive it as a low-priority issue. As a result, they do not support women seeking HCP services before pregnancy.

*Most of them (husbands) are obstacles, except a few* **(35 years old mother, not attended school, IDI**)

On the other hand, some participants highlighted that older mothers emphasize the importance of pre-pregnancy care, particularly for women facing infertility.

*"They are very supportive of it. They are especially encouraged if you give them recognition for mothers and children. It's okay because it's normal in the normal. However, women with the problem of being unable to get pregnant are encouraged to consult healthcare providers by the mothers because they want you to give birth to them. They even tell you to go in hiding"* **(Public health professional, male, KII)**

### High workload

HCPs noted that providing PCC services increases their workload and places additional burdens on staff, potentially leading to demands for extra benefits and staffing. When women come for child vaccinations and family planning, but also request PCC, it requires additional time and commitment from HCPs. Providers highlighted that due to time constraints, some women were not screened.

*"Implementing this program will increase our workload and place additional burdens on the staff, potentially leading to demands for extra benefits. This could pose a challenge if there is no allocated budget"* **(Midwifery professional, female, IDI)**

On the other hand, one HCP noted that if we work on PCC, it will make the provider's job easier because it decreases the risk factors like malnutrition, abortion, and other complications during pregnancy and childbirth.

*…."For sure, but if it is done here, I think it will make our job easier. So, we have worked on preconception care and are fixing this malnutrition. His silence, which we call abortion, is being fixed while it is there. Stunting and underweight are being fixed there. So, the professionals finish their work there"* **(Public health professional, male, KII)**

### Services cost

Low socioeconomic status, service costs, and transportation expenses pose major barriers to accessing PCC services. While maternal health services like antenatal care, labor, delivery, and postnatal care are free in Ethiopia, participants highlighted that the costs of PCC, such as lab tests and imaging, are challenging for low-income mothers. This financial burden can result in missed opportunities for PCC, increasing the risk of preventable maternal and child health complications. Additionally, many individuals prioritize others' needs over PCC if they feel healthy, and there is reluctance among mothers to seek care, even for routine checkups, when ill.

*"Regarding the affordability of PCC costs, a mother with a low income might not be able to afford to pay, especially for some lab tests and imaging procedures"* **(32 years old mother, 5th grade, IDI**)

*"If it has a cost, those who cannot afford it would miss the service, which could contribute to the occurrence of maternal and child health complications that could have been prevented otherwise "* **(29 years old mother, 10th grade, IDI**)

Mothers reported that long distances to health facilities and a lack of money for transportation often lead them to miss PCC services.

*"The health facility may be too far from your house, you may face a shortage of money, and you may be unable to walk there. In that case, how are you supposed to get to the health facility? If you do not have transportation costs, it is necessary to stay at home. Lack of money will prevent you from doing many things. Then, you become stressed and decide to be ignorant"* **(35 years old mother, not attended school, IDI**)

Nearly all mothers and HCPs agreed that PCC services should be accessible to individuals, regardless of economic status, to enhance maternal and child health, reduce costs, and promote equity. Participants stressed the importance of

providing these services free of charge, like other MCH programs, to support low-income mothers at higher risk of health complications.

*"The service should be free of charge. If it has a cost, those who cannot afford it will miss the service, which could contribute to the occurrence of maternal and child health complications that could have been prevented otherwise"* **(29 years old mother, 10ᵗʰ grade, IDI)**

The HCP working at MCH said that providing free PCC services is also difficult for the health facility, and balanced service payment may be good for sustaining the services

*"If it is a free service, the health facility may also suffer, but it should be at a reasonable price. For example, a sugar test results in 70 or 80 birrs. This should be reduced"* **(Public health professional, male, IDI)**

### Lack of medicines and medical equipment

Most of both HCPs and mothers highlighted that shortages of medical equipment and medicines such as iron and folic acid supplements, anti-hepatitis vaccines, immunoglobulin, laboratory reagents, essential drugs, and diagnostic tests like ultrasound pose significant challenges to PCC services, particularly at health centers. Similarly, they mentioned that due to the scarcity of medical supplies and medications, women are frequently referred to hospitals, exacerbating the situation, especially amid the crisis in the study area (Tigray). A participant working in the delivery unit highlighted a concerning trend of congenital anomalies, which they suspect may be linked to the shortage of folic acid supplements. Additionally, participants reiterated the challenges posed by the lack of guidelines and the shortage of HCPs

*"Most of the preconception care services, such as iron and folic acid supplements, ultrasounds, and lab tests for HIV/STI, are not available at the health center"* **(29 years old mother, 10ᵗʰ grade, IDI)**

4. Opportunities for PCC services

### Existing community platforms

Participants emphasized the crucial role of HEP in ongoing PCC awareness at the community level. They suggested enhancing this program by involving HEWs with WDGs and village health leaders. Participants also highlighted that home-to-home visits strengthen PCC, as some women may feel uncomfortable discussing their desire to conceive openly.

*"To make the program effective, it is essential to support WDGs and HEWs. In my previous work, I have seen the significant impact of WDGs in addressing public health issues. They have detailed community knowledge and can provide valuable information about vaccinated children, pregnant mothers, and other relevant data. By working with community leaders and WDGs, we can increase the acceptance and success of the program"* **(Midwifery professional, female, IDI)**

A Health extension worker said:

*"The home-to-home visit should be well strengthened because the woman might have felt ashamed to disclose her desire to get pregnant"* **(HEW, female, IDI)**

Participants suggested that the marriage certificate area could be an effective platform for PCC education and counseling. These venues offer couples an entry point for information and provide an opportunity to counsel them in advance, linking eligible women to health facilities.

*"Places where a marriage certificate is provided would be even better and of high importance to provide PCC service because it allows the couples to get the information before the start of sexual intercourse"* **(35 years old mother, 10ᵗʰ grade, IDI)**

Social networks such as civic societies, women's development groups, and farmers' associations are effective for community mobilization and spreading information. Participants highlighted their role in increasing PCC awareness and uptake through education and counseling at local venues, including churches, mosques, and traditional healing places.

*We can use the existing social networks such as civic societies (e.g., women's associations), women's development army, pregnant women forum, etc. Those social networks are the closest to the community; they can provide education to the mothers. First, the social networks should be aware of PCC and provide roles and responsibilities to teach and mobilize mothers to utilize the services. Social networks are very helpful in strengthening and decentralizing preconception care services* **(29 years old mother, 10th grade, IDI)**

SMART Start model: The SMART Start model is a girl-centered reproductive health initiative that promotes better pregnancy outcomes and well-being. It serves as an entry point for couples to address health risks by integrating contraceptive education, counseling, and financial planning, with a focus on achieving optimal health and readiness before pregnancy. Participants noted that this model highlights the importance of preparation for pregnancy, akin to pre-pregnancy care. To raise awareness and uptake of PCC, the services would be integrated into the SMART Start model program as an entry point for couples, particularly adolescent couples, through activities conducted by WDGs, HEWs, and other social networks.

*SMART Start is very good, especially because it advocates for adolescent girls to use contraceptives, and it creates awareness before pregnancy, so that we can use it. Similarly, as you have mentioned very well, women's development groups can be used as an entry pathway, as we know it is vital to be aware of adolescent girls regarding preconception care (Midwifery professional, male, KII)*

**Perceived desire for PCC services.** Though there would be a fear of workload, almost all HCPs appreciated the services benefit in preventing maternal and neonatal problems.

*…" There is also a higher need for PCC in the professional that can be described as an opportunity"* **(Public health professional, male, IDI)**

*Its necessity is unquestionable!* **(Public health professional, male, KII)**

In addition, many participants emphasized that, although these services are a new concept within the health system, mothers, especially those at high risk, have provided positive feedback.

*"If the service is available, plenty of mothers have a wish to get the service they desired"* **(35 years old mother, not attended school, IDI)**

*This is both appropriate and acceptable. Had education received to us in this form, we would not be afflicted by the illnesses and issues that we currently face. We'll say "Amen" to this, bring it down to our people, and persuade them to start working on it. That's a smart idea* **(39 years old mother, 3rd grade, FGD)**

5. Suggestions for PCC services

**Home-based initiative.** Participants pointed out that women intending to become pregnant need to prepare food and save money at home to ensure adequate nutrition, transportation, medications, and other necessities before and during pregnancy. Additionally, some participants emphasized the importance of communicating with their husbands to achieve a common understanding about the intention to get pregnant, being mentally prepared, and avoiding bad habits such as alcohol (including local beverages like "Siwa, Areki, and Myes"), smoking, and other addictive substances as pre-pregnancy care.

*Before her pregnancy, she should have a plan and be prepared economically; other work-related activities will help her overcome the problems that may arise following pregnancy.*

*In her home, she should communicate and have a common understanding with her husband about her pregnancy. When she plans to be pregnant, she should also avoid bad habits such as alcohol, smoking, and other addictive substances (hashish), and not be comfortable working during pregnancy in her home* **(Family health professional, female, KII)**

**Use various communication strategies.** All participants agreed that creating awareness for the infant program (PCC) through education on various platforms and social networks should be the first step to improving awareness of PCC. They emphasized the need for clear, culturally sensitive education on preparation, seeking advice, and understanding

preconception risks to address community beliefs, myths, and misconceptions. Utilizing media such as posters, billboards, and leaflets at community and facility levels is crucial for enhancing awareness. Once the community understands the importance of PCC, they will be more likely to seek it out independently.

*……" Anyway, they must work on awareness creation. Education on the causes and risk factors should be given. Education can be provided in the facilities, in groups, and in the community"* **(28 years old mother, 8th grade, IDI)**

*Any cultural or religious barriers are better addressed through education. We should educate the community in an understandable way. If we do so, I think the elders, community, and religious leaders would say yes, it is essential; there are only benefits, not harm; let's make use of it; and so on, if we educate them. I don't expect the other way around.* **(35 years old mother, 10th grade, IDI)**

The participants pointed out that education should be specific, pragmatic, and model based.

*Yes, of course: it is often better to see than hear to easily understand everything by looking at a picture. I think it would be better if he did the vegetables or other things diet in pictures* **(Midwifery professional, female, IDI)**

*If you give pretend or picture-based lessons or drama-assisted lessons, women understand easily* **(Public health professional, male, IDI)**

Participants suggested several strategies to increase awareness and PCC services, emphasizing the use of various media. Broadcast media like radio and television are particularly effective due to their widespread availability. Print media, including posters, billboards, and brochures, should be used at both community and facility levels. While HEWs play a crucial role, media influence is even more critical, as HEWs cannot reach every area. One participant noted that media can significantly raise awareness, stating that if people hear about PCC on the radio or television, they are more likely to seek health services and receive education through printed materials.

*"The greatest power is the media. Whether it is through television, radio, and magazines, the media is very crucial if you want to change the perception of the community about the PCC"* **(MD, male, IDI)**

*"Media has a big role. Our community follows the media. Although the role of HEWs is the biggest, the influence of media is more critical than anything because they can't reach every corner"* **(Nursing professional, female, IDI)**

**Community engagement.** Some participants emphasized that creating significant awareness about preconception care at the community level requires special efforts, mainly focusing on extensive information dissemination and community engagement.

*Special efforts are required, including robust community mobilization and active participation* **(HEW, female, IDI)**

**Community health facility Linkage.** In general, participants noted that establishing a linkage between the community and the health facility, as well as within the health facility to selected units providing PCC, using referral slips, would be vital for enhancing the uptake of PCC services and reducing delays in the health facility.

An MCH expert said *a referral slip is necessary. Once the eligible woman is identified, the referral slip can be used to link to the service. However, one thing that needs to be taken into consideration is whether it will be a free service. Maybe it could have some effect if it is a payable service* **(Public health professional, male, KII)**

**Use high-risk mothers as role model.** Some participants suggested that mothers with a history of APOs could serve as role models to educate women in women's development groups, pregnancy forums, and similar platforms. This approach could motivate the community to improve awareness and uptake of PCC.

*"I consumed three or four beers until I gave birth while I was pregnant. Here, I found myself teaching others and myself about mental retardation, a health issue that my child had developed. Even if you have to consider it before getting pregnant, you should not drink while pregnant. However, since my son experienced it, I have been telling people about it once more. Since a baby is like a piece of paper and can be easily injured, I think that my son's alcohol abuse contributed to his mental retardation"* **(27 years old mother, 10th grade, IDI)**

*"She also influenced other women who had difficulty conceiving, i.e., other women also seek health facilities after taking experience from my sister"* **(30 years old mother, diploma, FGD)**

## PCC services integration

HCPs unanimously recommended integrating PCC into existing services, including voluntary counseling and testing, family planning units, cervical cancer screening, youth-friendly services, ANC, ART clinics, post-abortion care, EPI, and under-five clinics. This approach aims to provide services to educate women of reproductive age who visit health facilities for other reasons, using the RLP tool to identify those eligible for PCC services effectively.

*"All people who visit a health facility should get information on PCC. We can use health facilities to disseminate information and create awareness"* **(HEW, Female, IDI)**

*"As a PCC, I believe that any female patient in the reproductive age group should be able to see you whenever they need something, like an abortion, an STI test, a family planning, a pregnancy test, or a PNC. Right now, all women should receive PCC counseling"* **(MD, male, IDI)**

Some participants suggested that PCC services should be provided as standalone services in a room, which would be more user-friendly. Integrating PCC with other services might make women uncomfortable. However, due to limited space, staff, and resources, offering PCC under the ANC unit could be beneficial, as it allows for the observation and advice on pregnancy-related abnormalities. Some participant suggests that PCC services could be offered alongside family planning services, where women typically seek to prevent pregnancy, providing an opportunity to discuss PCC.

## Discussion

This study examined the experiences, challenges, and opportunities of PCC services in northern Ethiopia following the government's prioritization of PCC as a strategic initiative and the introduction of new guidelines [6,7,26]. While previous qualitative research on PCC focused solely on barriers to its uptake [38], this study took a broader approach by exploring the awareness, experiences, challenges, and opportunities related to PCC services from the perspectives of various participants.

The study identified key themes related to PCC services, including awareness (information and perceived benefits) and experience (practices of PCC interventions and home-based preparation). Participants also explored perceived challenges, such as traditional beliefs and misconceptions, fragment-based service , reluctance to disclose conception intentions, societal norms, workload, and high costs. Additionally, they noted perceived opportunities, such as a strong interest in PCC services and the availability of community platforms. Suggestions for improvement included implementing home-based initiatives, adopting diverse communication strategies, involving high-risk mothers as role model, fostering community engagement, community-health facility linkage, and PCC services integration.

In the study, except for a few high-risk mothers, most mothers lack awareness of PCC services, which is consistent with previous studies [24,38–40]. Many people may be unfamiliar with PCC services because the program was introduced only recently [6,7,26] and has not yet been fully integrated into the healthcare system. This is especially evident within the health extension program, which is vital for increasing community awareness. This study highlights the issue, showing that HEWs possess less information about the program compared to other HCPs. Furthermore, HCPs often view PCC as an optional benefit rather than a standard service, leading to limited investment and delayed care-seeking until pregnancy is confirmed. Such neglect can hinder health-seeking behavior and the utilization of services. Overall, participants emphasized that enhancing PCC awareness should be a government priority, leveraging community engagement through social networks like civic organizations, women's groups, and farmers' associations, which are effective for mobilization and information sharing. Home-based preparation was identified as a promising foundation, alongside diverse communication methods and integrating PCC services into healthcare facilities to improve awareness and utilization. Evidence also showed that to enhance PCC awareness, massive public awareness campaigns and education through print media, social forums, and government-led initiatives, including integrating PCC into the healthcare services [22,41]. On the other hand, approximately 76% of women were aware of PCC [42]. This difference may be attributed to variations in service accessibility, health-seeking behavior, and educational levels.

This study revealed that while most HCPs had some knowledge of PCC services, their practice was limited. The services mainly focused on specific components targeting high-risk mothers, such as contraceptive counseling for pregnancy delay, medication safety, substance use counseling, and folic acid supplementation. However, most interventions recommended by the WHO and Ethiopia's national PCC guideline were not practiced [2,7]. This is consistent with previous findings from African countries, where PCC was implemented under inaccessible guidelines and primarily driven by initiatives targeting high-risk women with a limited focus on interventions such as dietary modification counseling [25,43,44]. In contrast, approximately 88% of HCPs in South Africa report practicing PCC [45]. The limited practice of PCC is influenced by several factors, such as the presence of a relatively new program, HCPs' lack of adequate training, and PCC guidelines not being easily accessible. Moreover, the government prioritizes ANC over PCC, leading HCPs to view PCC as a supplementary rather than a routine service. Evidence suggests that accessible guidelines and regular use of RLP tools can help HCPs prioritize PCC and emphasize preconception care for women [22,25]. Hence, prioritizing PCC and integrating package-based services into routine care for all women of reproductive age using RLP tools could significantly enhance awareness and utilization.

The study revealed that traditional beliefs often compel women to keep their conception desires private, avoiding discussions about PCC. Pregnancy is commonly regarded as a divine gift from St. Mary or God, and many women believe that expressing a desire to conceive before pregnancy contradicts divine will, making such discussions socially and personally taboo. This is consistent with other studies elsewhere [38,46], indicating that women often perceive conception as a natural event requiring no preparation, resulting in limited communication with HCPs and missed opportunities to utilize PCC services. Evidence suggests that addressing traditional beliefs about conception requires collaboration between health institutions and religious organizations [38]. Additionally, integrating preconception health discussions into routine healthcare, beginning with school-based services, has been proposed as a potential solution [47]. In this study, participants indicated that women prefer receiving PCC at health posts and health centers, as well as through community-based services, because these settings align with cultural norms and are conveniently located, making it easier for women to discuss personal matters.

Evidence suggests that without adequate counseling, clients frequently seek information from friends and family sources that often provide inaccurate guidance [48]. Common misconceptions, such as the belief that contraceptives like Depo-Provera lead to infertility or birth defects, deter their use and impede the adoption of PCC. Participants in this study identified insufficient counseling within contraceptive services as a major barrier to PCC awareness and utilization, aligning with findings from another research. While HEWs prioritize maternal care, they often overlook PCC as part of a comprehensive service package, representing a significantly missed opportunity. Additionally, evidence underscores the vital role of family planning units in enhancing preconception health by assessing pregnancy intentions and dispelling misconceptions through tools like the RLP [49]. Therefore, integrating PCC into family planning and antenatal services strengthens the HEP with culturally sensitive education, and utilizing health visits for PCC education can significantly enhance services and reduce misconceptions about contraceptive use.

Involving husbands in PCC through screenings and counseling is vital for reducing risks and promoting healthier pregnancies, as emphasized by Ethiopian [7] and Chinese national guidelines [50]. Furthermore, women who made decisions independently or in partnership with their husbands were more likely to utilize PCC services [51,52]. Our findings showed that husbands often oppose women's PCC, particularly in rural and remote areas. Poor parental behaviors before conception are linked to increased illness and mortality in offspring, while healthier habits can significantly improve pregnancy outcomes. Paternal preconception care emphasizes men's direct contributions to child health, such as genetic and epigenetic factors, lifestyle choices, environmental exposures, and indirect influences through partner health and relationships [53]. The paternal origins of the health and disease model define [54] the preconception population as all reproductive-age men and women, highlighting the importance of paternal health. Given men's crucial role in reproductive health decisions, couple-based counseling is strongly recommended [48].

The study found that poor socioeconomic status, including service costs and distance, hinders PCC use, with some viewing pregnancy as a luxury due to basic needs and regional conflicts. These findings align with results from various other studies [38,55]. The implication is that even when women acknowledge the need for PCC and wish to access it, they face additional challenges, such as service charges and transportation costs. Participants suggested women prefer receiving PCC at health posts and health centers alongside community-based services. They feel more comfortable discussing personal matters in culturally familiar local settings rather than hospitals. This preference helps address transportation costs, offers more apparent education, and overcomes barriers related to disclosing pregnancy intentions, which is a major obstacle to PCC utilization. Besides, the current study emphasizes that HCPs and mothers strongly advocate for making PCC services free, as they are part of the maternal continuum of care. This could improve access, particularly for low-income mothers, enhance maternal and child health, reduce costs, and promote equity. Other study in China support this, with China's National Free Preconception Health initiative, for example, reaching over 95% of target couples with preconception health education [50]. Additionally, free maternal healthcare, including PCC, aligns with WHO's recommendation to eliminate financial barriers and ensure equitable access to healthcare for all. It supports achieving SDG 3.1, which aims to reduce maternal mortality to fewer than 70 per 100,000 live births by 2030 [56].

To enhance awareness and uptake of PCC, the study suggests using marriage certificate settings to provide couples with information and pre-counseling, thereby connecting eligible women to health facilities. This approach could be practical if the Ethiopian government, in collaboration with religious leaders, commits to issuing marriage certificates, particularly for newly married couples. This recommendation aligns with findings from a study in Bangladesh [57]. Marriage certificate settings present an ideal opportunity to introduce PCC services, as couples are often in the early stages of planning their future. This universal process provides an equitable platform to educate couples on reproductive health, family planning, genetic risks, and the importance of preparing for a healthy pregnancy.

According to the Motivational Theory of Role Modeling, role model-based interventions are highly effective in encouraging others to accept new services [58]. In our study, participants suggested that high-risk mothers could act as role models to enhance awareness and encourage the use of PCC services.

Ethiopia has integrated PCC into its reproductive health strategy (2020–2024/25) [26], and ANC guidelines (2022) [6], with recent 2024 guidelines recommending services for women planning pregnancy within three months [7]. However, implementation remains weak due to a lack of standardized guidelines, insufficient government prioritization, inadequate training, and fragmented delivery that depends on women's self-initiation and is rarely offered proactively, even to high-risk women or those discontinuing contraception to conceive get challenges seen in other African studies [44,59]. In contrast, studies [50,60] indicated that fully implemented services supported by established protocols and strong government backing, improve service utilization compared to fragmented approaches. To improve uptake, study participants suggested integrating PCC into existing services such as ANC, HIV testing, ART clinics, and family planning using the RLP tool for women of reproductive age. This aligns with other studies in Africa [4,5]. An integrated approach systematically delivers PCC to all reproductive-aged women via the RLP tool, overcoming barriers of low awareness, passive pregnancy planning, and infrequent healthcare use demonstrably improving awareness and uptake [22,25].

## Study limitations and strengths

This first study in Ethiopia explores PCC services from multiple perspectives, including front-line HCPs in urban and rural areas. The findings significantly contribute to developing locally appropriate programs to increase awareness of PCC and uptake. However, due to resource constraints, our study did not explore the perspectives of husbands, religious leaders, and women without pregnancy histories, emphasizing the need for further research. We held open discussions and regular meetings among the researchers to maintain trustworthiness. However, because the naming of the service is unfamiliar to the community, some women may have confused PCC with ANC, potentially leading to exaggerated responses.

## Conclusions

This study reveals that most women lack awareness of PCC and that HCPs currently deliver fragmented PCC services without standardized guidelines or assessment tools, primarily focusing on high-risk cases. Key challenges identified include service fragmentation, traditional beliefs and misconceptions, inadequate counseling on contraceptive services, and strong social influences. Despite these challenges, there are opportunities to strengthen PCC through existing community platforms, particularly HEP. To enhance the PCC services, the study recommends integrating services into community health systems and employing diverse communication strategies such as media campaigns and educational programs to dispel misconceptions and raise awareness. Furthermore, strengthening home-based initiatives and engaging high-risk mothers as community role models may effectively promote PCC. A package-based approach should also be adopted by introducing the concept of RLP and enabling the routine use of RLP tools to identify eligible women for PCC services. These findings provide critical insights for Ethiopia's Ministry of Health, offering guidance to policymakers and program designers aiming to improve PCC uptake. Future interventions should prioritize comprehensive service packages, demand creation, and stronger service linkages to optimize PCC implementation.

## Supporting information

**S1 File. COREQ Checklist for Study Reporting.**
(PDF)

**S2 File. Qualitative Guide for Preconception Care Services.**
(DOCX)

## Author contributions

**Conceptualization:** Gebremedhin Gebreegziabher Gebretsadik, Andargachew Kassa Biratu, Zohra S. Lassi, Afework Mulugeta.

**Data curation:** Gebremedhin Gebreegziabher Gebretsadik.

**Formal analysis:** Gebremedhin Gebreegziabher Gebretsadik, Afework Mulugeta.

**Funding acquisition:** Gebremedhin Gebreegziabher Gebretsadik.

**Investigation:** Gebremedhin Gebreegziabher Gebretsadik, Afework Mulugeta.

**Methodology:** Gebremedhin Gebreegziabher Gebretsadik, Andargachew Kassa Biratu, Amanuel Gessessew, Afework Mulugeta.

**Project administration:** Gebremedhin Gebreegziabher Gebretsadik, Afework Mulugeta.

**Resources:** Gebremedhin Gebreegziabher Gebretsadik, Afework Mulugeta.

**Software:** Gebremedhin Gebreegziabher Gebretsadik, Afework Mulugeta.

**Supervision:** Gebremedhin Gebreegziabher Gebretsadik, Alemayehu Bayray Kahsay, Andargachew Kassa Biratu, Amanuel Gessessew, Zohra S. Lassi, Afework Mulugeta.

**Validation:** Gebremedhin Gebreegziabher Gebretsadik, Alemayehu Bayray Kahsay, Andargachew Kassa Biratu, Amanuel Gessessew, Zohra S. Lassi, Afework Mulugeta.

**Visualization:** Gebremedhin Gebreegziabher Gebretsadik, Alemayehu Bayray Kahsay, Andargachew Kassa Biratu, Amanuel Gessessew, Zohra S. Lassi, Afework Mulugeta.

**Writing – original draft:** Gebremedhin Gebreegziabher Gebretsadik.

**Writing – review & editing:** Gebremedhin Gebreegziabher Gebretsadik, Alemayehu Bayray Kahsay, Andargachew Kassa Biratu, Amanuel Gessessew, Zohra S. Lassi, Afework Mulugeta.

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
