## [Decision Letter · Decision Letter 0]

2 Dec 2024

Dear Dr. Gebretsadik,

Thank you for submitting your manuscript to PLOS ONE. After careful consideration, we feel that it has merit but does not fully meet PLOS ONE’s publication criteria as it currently stands. Therefore, we invite you to submit a revised version of the manuscript that addresses the points raised during the review process.

We look forward to receiving your revised manuscript.

Kind regards,

Marianne Clemence

Staff Editor

PLOS ONE

Journal Requirements:

3. In the ethics statement in the Methods, you have specified that verbal consent was obtained. Please provide additional details regarding how this consent was documented and witnessed, and state whether this was approved by the IRB

4. We note that your Data Availability Statement is currently as follows: “All relevant data are within the manuscript and in Supporting Information files.”

Please confirm at this time whether or not your submission contains all raw data required to replicate the results of your study. Authors must share the “minimal data set” for their submission. PLOS defines the minimal data set to consist of the data required to replicate all study findings reported in the article, as well as related metadata and methods (https://journals.plos.org/plosone/s/data-availability#loc-minimal-data-set-definition). For example, authors should submit the following data: - The values behind the means, standard deviations and other measures reported; - The values used to build graphs; - The points extracted from images for analysis. Authors do not need to submit their entire data set if only a portion of the data was used in the reported study. If your submission does not contain these data, please either upload them as Supporting Information files or deposit them to a stable, public repository and provide us with the relevant URLs, DOIs, or accession numbers. For a list of recommended repositories, please see https://journals.plos.org/plosone/s/recommended-repositories. If there are ethical or legal restrictions on sharing a de-identified data set, please explain them in detail (e.g., data contain potentially sensitive information, data are owned by a third-party organization, etc.) and who has imposed them (e.g., an ethics committee). Please also provide contact information for a data access committee, ethics committee, or other institutional body to which data requests may be sent. If data are owned by a third party, please indicate how others may request data access.

Reviewers' comments:

Reviewer's Responses to Questions

**Comments to the Author**

1. Is the manuscript technically sound, and do the data support the conclusions?

Reviewer #1: Yes

Reviewer #2: Partly

2. Has the statistical analysis been performed appropriately and rigorously?

Reviewer #1: No

Reviewer #2: N/A

3. Have the authors made all data underlying the findings in their manuscript fully available?

Reviewer #1: Yes

Reviewer #2: Yes

4. Is the manuscript presented in an intelligible fashion and written in standard English?

Reviewer #1: Yes

Reviewer #2: No

Reviewer #1: The findings section requires a complete overhaul. Most of the tables are not clear. In Table 2, in the narratives, you mentioned 5 themes, but the table contains only 4 themes. In the same vein, you mentioned 29 subthemes, but the table contains 30 subthemes

Reviewer #2: PONE-D-24-41198_Review comments

General comments

There is a need for English editing of the paper to correct the grammatical errors and improve the clarity of the presentation.

Title

Policy Implications of Preconception Care Services: Experiences, Challenges, and Opportunities in Tigray, Northern Ethiopia; an Exploratory Qualitative Study

I suggest removing “policy implications” from the title as the manuscript does not speak to policy clearly. The current title does not align well with the stated aim of the study to explore experiences, challenges and opportunities. I suggest something along these lines “Awareness and uptake of preconception care services in Tigray, Northern Ethiopia – a qualitative exploration of experiences, challenges and opportunities”.

Methods

Study design

1. The sentence “Since 2020, Ethiopia has strategically integrated PCC into its health system, guided by newly developed guidelines (6, 7).” will fit better in the introduction rather than in the study setting. If available data on the service delivery so far can be provided.

2. “With” needs to be deleted in this statement. “These zones include 591,481 women of reproductive age (23.5% of the population) and employ 1,731 healthcare providers, including 350 with Health extension workers (HEWs).”

3. The reference to a war in the statement “However, the war damaged over 80% of health facilities, leading to a 40% decline in maternal and child health services, including institutional deliveries (26).” needs to be clarified better. It may be useful to include a paragraph in the introduction describing the state of health services before and after the war including dates and possible reasons for the unrest. This will enable readers to understand the context better.

Recruitment of participants

1. The statement “We identified participants communicated with HEWs and women development group (WDGs) from HEW registers using purposive sampling, considering their pregnancy and risks.” is unclear; the sentence appears incomplete.

2. How did the authors determine the “intention to become pregnant”? Was there any screening questionnaire? Where were potential participants identified? The health facility or within the community? Who identified the participants in either instance?

3. Similarly, how did the authors identify the women who had a history of adverse pregnancy outcomes?

4. What do ESOGA & EMA mean?

5. What strategies did the research team apply to “bracket” their prior experiences?

Ethics approval and consent to participate

The statement “Before data collection, we attached a one-page consent form to the questionnaire, explaining participants' autonomy” implies that a survey was conducted whereas the study is described as qualitative. Was there a questionnaire survey in addition to the qualitative data collection? The methods need to be clarified appropriately.

Results

1. What is the justification for including teenagers in the study? Table 1 shows that there are two intending mothers between 15 and 19 years old.

2. Figure 1 – conceptual framework: please provide a brief description of the relationships between the concepts in the framework. A legend describing the directions of the different arrows included will also be helpful.

3. The triangulation of results can be improved. Some sections are fairly clear with opposing ideas around the same theme well presented. In other sections the presentation is not well aligned. It will also be useful to compare the opinions of health care providers and the different groups of women on the points raised to make the discussion more robust.

4. In some places the authors have written St Mary and in others St Merry. If this is a reference to the same religious figure the spellings should be aligned.

Discussion

The “SMART Start model” is mentioned in the results and again in the discussion. Is this a model described in the Ethiopian health care system? It would be useful to provide some explanation of the model and proffer possible suggestions on why it was mentioned as a potential strategy for delivery of PCC care in the study.

It would be useful for the authors to proffer possible solutions to the challenges to PCC service provision and uptake identified in the study rather than only restate the issues already highlighted in the results. Some of the themes in the results can be rephrased as strategies and opportunities for PCC services.

**Do you want your identity to be public for this peer review?** For information about this choice, including consent withdrawal, please see our Privacy Policy

Reviewer #1: **Yes: ** Winifred Chinyere Ukoha

Reviewer #2: No

---

## [Author Response · Author response to Decision Letter 1]

18 Jan 2025

we respond your comments and suggestion point by point. the detailed description was included in the respond to review document

---

## [Decision Letter · Decision Letter 1]

23 Apr 2025

Dear Dr. Gebretsadik,

Thank you for submitting your manuscript to PLOS ONE. After careful consideration, we feel that it has merit but does not fully meet PLOS ONE’s publication criteria as it currently stands. Therefore, we invite you to submit a revised version of the manuscript that addresses the points raised during the review process.

**Please note that the reviewers have noted a number of suggestions from the previous round of revision that have not been sufficiently addressed. Please take care to address them in this current revision, with particular attention to methodological/analyses details.**

We look forward to receiving your revised manuscript.

Kind regards,

Avanti Dey, PhD

Staff Editor

PLOS ONE

**Journal Requirements:**

Reviewers' comments:

Reviewer's Responses to Questions

**Comments to the Author**

Reviewer #1: (No Response)

Reviewer #2: (No Response)

2. Is the manuscript technically sound, and do the data support the conclusions?

Reviewer #1: Yes

Reviewer #2: Yes

3. Has the statistical analysis been performed appropriately and rigorously?

Reviewer #1: Yes

Reviewer #2: N/A

4. Have the authors made all data underlying the findings in their manuscript fully available?

Reviewer #1: No

Reviewer #2: Yes

5. Is the manuscript presented in an intelligible fashion and written in standard English?

Reviewer #1: Yes

Reviewer #2: Yes

**Reviewer #1: ** The authors should attach the COREQ checklist

The discussion section is a bit too long and can be shortened

What do you mean by "topic guide"? Line 190

Reprase the last subtheme

The authors still did not provide any justifications for some of their findings

**Reviewer #2: ** The authors have revised the manuscript adequately, however some of the issues raised in the previous review were addressed in the review table but not inserted into the manuscript. The authors need to check that these revisions are made on the manuscript as appropriate.

**Do you want your identity to be public for this peer review?** For information about this choice, including consent withdrawal, please see our Privacy Policy

Reviewer #1: No

Reviewer #2: No

---

## [Author Response · Author response to Decision Letter 2]

29 Apr 2025

We have addressed the editors’ and reviewers’ comments point by point, and we would be honored to have our manuscript accepted for publication in your journal.

---

## [Decision Letter · Decision Letter 2]

5 Nov 2025

Dear Dr. Gebretsadik,

Thank you for submitting your manuscript to PLOS ONE. After careful consideration, we feel that it has merit but does not fully meet PLOS ONE’s publication criteria as it currently stands. Therefore, we invite you to submit a revised version of the manuscript that addresses the points raised during the review process.

We look forward to receiving your revised manuscript.

Kind regards,

Wen-Jun Tu

Academic Editor

PLOS ONE

Journal Requirements:

Reviewers' comments:

Reviewer's Responses to Questions

**Comments to the Author**

Reviewer #1: (No Response)

2. Is the manuscript technically sound, and do the data support the conclusions?

Reviewer #1: Yes

3. Has the statistical analysis been performed appropriately and rigorously?

Reviewer #1: N/A

4. Have the authors made all data underlying the findings in their manuscript fully available?

Reviewer #1: Yes

5. Is the manuscript presented in an intelligible fashion and written in standard English?

Reviewer #1: Yes

Reviewer #1: Thanks to the authors for improving the paper by implementing the previous comments. Receive the additional comments below, which may assist in strengthening the paper further.

**Do you want your identity to be public for this peer review?** For information about this choice, including consent withdrawal, please see our Privacy Policy

Reviewer #1: No

---

## [Author Response · Author response to Decision Letter 3]

8 Nov 2025

We have addressed all points raised by the editors and reviewers in a point-by-point response. We believe the manuscript is now suitable for publication and will provide critical foundational evidence for shaping infant health programs in Ethiopia.

---

## [Editor Report · Decision Letter 3]

10 Nov 2025

Preconception Care Services in Northern Ethiopia: A Qualitative Exploration of Awareness, Experiences, Challenges, Opportunities, and Prospects

PONE-D-24-41198R3

Dear Dr. Gebretsadik,

We’re pleased to inform you that your manuscript has been judged scientifically suitable for publication and will be formally accepted for publication once it meets all outstanding technical requirements.

Kind regards,

Wen-Jun Tu

Academic Editor

PLOS ONE
---

## [Editor Report · Acceptance letter]

PONE-D-24-41198R3

PLOS ONE

Dear Dr. Gebretsadik,

I'm pleased to inform you that your manuscript has been deemed suitable for publication in PLOS ONE. Congratulations! Your manuscript is now being handed over to our production team.

Kind regards,

on behalf of

Dr. Wen-Jun Tu

Academic Editor

PLOS ONE